# Aged G Protein-Coupled Receptor Kinase 3 (Grk3)-Deficient Mice Exhibit Enhanced Osteoclastogenesis and Develop Bone Lesions Analogous to Human Paget’s Disease of Bone

**DOI:** 10.3390/cells12070981

**Published:** 2023-03-23

**Authors:** Emily M. Rabjohns, Rishi R. Rampersad, Arin Ghosh, Katlyn Hurst, Amanda M. Eudy, Jaime M. Brozowski, Hyun Ho Lee, Yinshi Ren, Anthony Mirando, Justin Gladman, Jessica L. Bowser, Kathryn Berg, Sachin Wani, Stuart H. Ralston, Matthew J. Hilton, Teresa K. Tarrant

**Affiliations:** 1Division of Rheumatology and Immunology, Duke University Department of Medicine, Durham, NC 27710, USA; 2Department of Pathology and Laboratory Medicine, University of North Carolina at Chapel Hill, Chapel Hill, NC 27599, USA; 3College of Arts and Sciences, Duke University, Durham, NC 27510, USA; 4Department of Orthopaedic Surgery, University of Texas Southwestern, Dallas, TX 75390, USA; 5Scottish Rite Hospital, Dallas, TX 75219, USA; 6Department of Orthopedics, Duke University, Durham, NC 27710, USA; 7Pratt School of Engineering, Duke University, Durham, NC 27710, USA; 8Lineberger Comprehensive Cancer Center, University of North Carolina at Chapel Hill, Chapel Hill, NC 27599, USA; 9Centre for Genomic and Experimental Medicine, University of Edinburgh, Edinburgh EH4 2XU, UK; 10Durham Veterans Hospital, Durham, NC 27710, USA

**Keywords:** bone, osteoclast, Paget’s, Paget’s disease of bone, G protein-coupled receptor kinase, G protein-coupled receptor, metabolic bone disease

## Abstract

Paget’s Disease of Bone (PDB) is a metabolic bone disease that is characterized by dysregulated osteoclast function leading to focal abnormalities of bone remodeling. It can lead to pain, fracture, and bone deformity. G protein-coupled receptor kinase 3 (GRK3) is an important negative regulator of G protein-coupled receptor (GPCR) signaling. GRK3 is known to regulate GPCR function in osteoblasts and preosteoblasts, but its regulatory function in osteoclasts is not well defined. Here, we report that *Grk3* expression increases during osteoclast differentiation in both human and mouse primary cells and established cell lines. We also show that aged mice deficient in *Grk3* develop bone lesions similar to those seen in human PDB and other Paget’s Disease mouse models. We show that a deficiency in *Grk3* expression enhances osteoclastogenesis in vitro and proliferation of hematopoietic osteoclast precursors in vivo but does not affect the osteoclast-mediated bone resorption function or cellular senescence pathway. Notably, we also observe decreased *Grk3* expression in peripheral blood mononuclear cells of patients with PDB compared with age- and gender-matched healthy controls. Our data suggest that GRK3 has relevance to the regulation of osteoclast differentiation and that it may have relevance to the pathogenesis of PDB and other metabolic bone diseases associated with osteoclast activation.

## 1. Introduction

Paget’s Disease of Bone (PDB) is the second most common metabolic bone disorder, affecting about 0.3% of patients older than 55 in the UK. It is characterized by focal bone lesions that can be associated with deformity, pain, pathologic fracture, and/or nerve compression [1,2]. Mechanistically, the lesions are caused by an imbalance between osteoclast and osteoblast activity [1,2,3]. Osteoclasts are formed by the fusion of myeloid cell precursors to form large multinucleated cells that adhere to the bone surface and remove bone mineral by secreting hydrochloric acid and cathepsin K. It is thought that osteoclast overactivity is central to the pathogenesis of PDB, leading to osteolytic and osteosclerotic focal lesions with formation of woven bone with a disorganized structure, that reduces the mechanical strength of bone [1,2,3].

Several genes have been implicated in the pathogenesis of PDB, particularly in familial kindreds. In 2002, mutations in the *SQSTM1* gene were identified as being causative in a proportion of adult-onset PDB cases [4]. *SQSTM1* encodes p62, a scaffolding protein important for several signaling pathways and cellular functions, including facilitating ubiquitination, regulating autophagy, and apoptosis [5,6]. Most mutations in *SQSTM1* associated with PDB have been localized to the ubiquitin-binding domain [5,6]. A Genome Wide Association Study (GWAS) identified loci associated with several genes previously known to be involved in osteoclastogenesis, including *TNFRSF11A*, *CSF1*, and *DC-STAMP*, as well as *OPTN* [7]. Other PDB genotype–phenotype association studies have found osteoprotegerin (*TNFRSF11B*) polymorphisms with increased prevalence in both familial and non-familial PDB cases [8]. Separate analyses suggest that polymorphisms in osteoprotegerin in PDB are sex-specific [9].

Mouse models of PDB have been generated based on the above-mentioned studies of human genetic risk loci. One mouse model engineered to express a *p62^P392L^* mutation under an osteoclast-specific promoter did not recapitulate a full PDB phenotype. Cells from these mice displayed enhanced osteoclastogenesis and the mice had low bone volume, but no focal bone lesions were observed in mice aged up to 18 months [10]. However, global expression of a *p62^P394L^* mutation generated a mouse model that exhibits disease similar to PDB with skeletal lesions appearing around 8 months of age and progressing in severity with advancing age [11]. Further studies show that treatment with zoledronic acid, a bisphosphonate used to treat PDB, was effective at preventing the formation of lesions in the *p62^P394L^* model [12].

Mice with loss of function in the optineurin autophagy receptor *OPTN^D477N/D477N^* exhibit altered bone turnover but do not readily develop pagetic bone lesions; only ~10% of the cohort aged 15–18 months developed a bone lesion. Additionally, mice with a truncating mutation in *OPTN* (*OPTN^ΔE×12/ΔE×12^*) did not develop bone lesions [13]; however, global deletion of the optineurin protein (*Optn^−/−^*) recapitulates a PDB-like phenotype, with osteolytic and mixed osteolytic–osteosclerotic lesions occurring with advanced age (16 months) and age-dependent elevation in serum alkaline phosphatase (ALP) [14]. Loss of *OPTN* expression results in impaired IFNβ production and signaling downstream of IFNα/βR, resulting in enhanced osteoclastogenesis via a loss of inhibitory factors [14]. More recently, *Optn^−/−^* mice have been shown to have increased intracellular reactive oxygen species with increased oxidative stress in tissues that may be contributing to the enhanced osteoclastogenesis [15].

Signaling through G protein-coupled receptors (GPCR) is ubiquitous in biology and has relevance to bone homeostasis and pathophysiology. GPCR signaling is negatively regulated by G protein receptor kinases (GRKs), which are expressed in several cell subtypes within the bone marrow niche [16,17]. Parathyroid hormone receptor (PTH-R) signaling in osteoblasts is primarily regulated by GRK2 as opposed to other GRK subtypes [18,19,20]. Our recent work shows that *Grk3*-deficient (*Grk3^−/−^)* bone marrow-derived mesenchymal stem cells (BmMSCs) have augmented osteogenic differentiation and proliferation regulated in part through sphingosine-1-phosphate receptor (S1PR) signaling [21]. Given the proliferative osteoblastic phenotype that we observed in BmMSCs in vitro, we hypothesized that the *Grk3^−/−^* mouse would have a protective phenotype against osteoporosis and preserved bone density in aging; however, this was surprisingly not observed [21]. Upon further investigation, we found that aged *Grk3^−/−^* mice had cortical bone density irregularities on µCT and histopathologic and radiographic lesions similar to those seen in PDB patients and in mouse models of PDB. This report describes our discovery of aged mice deficient in GRK3 that share similarities to PDB that could be either diagnostically or therapeutically important to human metabolic bone disease.

## 2. Materials and Methods

### 2.1. Query of Publicly Available Expression Data

Datasets were queried and visualized using Genevestigator and GEO2R applications. Queried datasets include GSE107295, GSE10246, GSE63009, GSE30160, and GSE43811.

### 2.2. Mice

The *Grk3^−/−^* mouse strain was kindly provided by Dr. Robert J. Lefkowitz of Duke University and backcrossed > 12 generations on the C57BL/6 background. The line is rederived every 1–2 years to prevent genetic drift from the C57BL/6 strain. Animals were housed in standard IACUC-approved housing conditions under the care of Duke University’s Division of Laboratory Animal Resources under protocol #A154-20-07.

### 2.3. Grk3 Knockdown of RAW 264.7 Cells

RAW 264.7 cells (hereafter termed “RAW” cells) were purchased from American Type Culture Collection (ATCC^®^ TIB71™). RAW cells were cultured in “RAW culture media”, DMEM (Sigma-Aldrich, St. Louis, MO, USA) supplemented with 10% fetal bovine serum (“FBS”) (Atlanta Biologicals, Flowery Branch, GA, USA) and 1% penicillin/streptomycin (Corning, NY, USA). Cells were plated at 1.6 × 10^4^ cells/well in a 96-well plate. A titration (1 µL, 2 µL and 5 µL) of Sigma MISSION shRNA Lentiviral Transduction Particles (Sigma, St. Louis, MO, USA) containing either non-target control shRNA, or *Grk3*-specific shRNA (TRCN0000022700 and TRCN0000022701) was introduced to RAW cells and incubated overnight in media containing 8 µg/mL hexadimethrine bromide (Sigma-Aldrich, St. Louis, MO, USA). The following day, 4 µg/mL of puromycin was introduced and successful transfection was demonstrated by puromycin resistance. Nomenclature for the RAW *Grk3*-knockdown clones (“RAW *Grk3*-KD”) hereafter is the final three digits of the shRNA clone number and the MOI of 2.

### 2.4. Osteoclast Differentiation of RAW Cells

Transfected cells were plated at 4 × 10^3^ cells/well in a 12 well plate and differentiated in DMEM (Sigma, St. Louis, MO, USA) supplemented with 10% FBS, 1% penicillin/streptomycin, 4 µg/mL puromycin (Sigma, St. Louis, MO, USA) and 50 ng/mL Receptor Activator of Nuclear Factor κB (RANKL) (Invitrogen eBioscience, Waltham, MA, USA) for five days.

### 2.5. Grk3 mRNA Expression of RAW Cells, RAW-OCLs, and Grk3-Knockdown RAW Clones

mRNA from undifferentiated and RANKL-differentiated RAW *Grk3*-KD clones and the control clone was prepared using the RNeasy Plus Mini Kit (Qiagen, Germantown, MD, USA) according to manufacturer’s instructions. Reverse transcriptase cDNA synthesis was performed on 100 ng mRNA template using SuperScript IV VILO Master Mix (Invitrogen, Waltham, MA, USA). qRT-PCR was performed in duplicate (SYBR^®^ Green, Bio-Rad, Hercules, CA, USA) and normalized to housekeeping gene *IDUA*. Mean fold change of *Grk3* was determined by 2-ΔΔCt with NT-2 as control. Primers utilized for qRT-PCR were *Grk3* forward: AAG CCT TCG TGG GGA TAT TT, *Grk3* reverse: TCG TTC ATG CTC AGG TGA AT; *IDUA* forward: GCA TCC AAG TGG GTG AAG TT and *IDUA* reverse: CAT TGA GCA GGT CCG GAT AC.

### 2.6. Flow Cytometric Analysis of RAW Grk3-Knockdown Osteoclasts

RAW *Grk3*-KD clones 700-2 and 701-2 and the NT-2 control clone were cultured in T75 flasks until 70–80% confluent. Cells were harvested using cell scrapers and seeded in 24-well plates at 3 × 10^4^ cells per well. RAW culture media was supplemented with RANKL (50 ng/mL Invitrogen eBioscience, MA, USA, or 20 ng/mL Biolegend, San Diego, CA, USA) at plating and media was replaced daily. To harvest cells for flow cytometry, media was aspirated, wells were washed and 500 µL of Accutase (Sigma, St. Louis, MO, USA) was added and incubated at 37 °C for 30–40 min. Cells were collected and stained in PBS supplemented with 1% FBS and 2 mM EDTA (VWR, Suwanee, GA, USA). Cell viability was ascertained by trypan blue exclusion. H33342 nuclear stain (Sigma, St. Louis, MO, USA) was added to collection tubes at 5 µg/mL and incubated at 37 °C for 60 min. Cells were diluted with 500 µL ice cold staining buffer, spun, and resuspended in 100 µL ice cold staining buffer with 1U DNase I (Invitrogen, Waltham, MA, USA) for flow cytometry. A low flow rate (up to 2000 events per second) was maintained throughout. Flow cytometry was analyzed on FlowJo v10 software (Becton Dickinson, Ashland, OR, USA). Gating strategy was based on the procedure described by Madel, et al. [22].

### 2.7. Resorption Assay

Bone resorption was measured using a fluoresceinamine-labeled chondroitin sulfate (FACS)-coated, 48-well bone resorption assay kit (Cosmo Bio, Carlsbad, CA, USA) plate. RAW *Grk3*-KD clones 700-2 and 701-2 and the NT-2 control clones were seeded at 5 × 10^3^ cells per well in phenol red free DMEM (Thermo Fisher, Waltham, MA, USA) containing 10% FBS, 1% L-glutamine, 1% penicillin/streptomycin, 4 µg/mL puromycin, and supplemented either with or without 20 ng/mL RANKL (Biolegend, San Diego, CA, USA). At days 3, 5, and 7 the conditioned media was transferred to a 96-well black plate, mixed with bone resorption buffer, and fluorescence immediately measured with a fluorometric plate reader using an excitation wavelength of 485 nm and emission at 535 nm. Net fluorescence was obtained by subtracting the values of the undifferentiated cells.

### 2.8. Analysis of Primary Wild-Type and Grk3-Deficient Osteoclasts

Wild-type (WT) and *Grk3*^−/−^ mice were euthanized using CO_2_ followed by secondary confirmation as per Duke University IACUC-approved methods. The carcass was soaked in 70% ethanol and the femur and tibia were dissected out and flushed using cold PBS supplemented with 10% FBS using 27 G needles. The bone marrow single-cell suspension was filtered through a 70 µM cell strainer (Fisher Scientific, Waltham, MA, USA) and RBC lysis was performed using Hybri-Max Red Blood Cell Lysing Buffer (Sigma-Aldrich, St. Louis, MO, USA) as per the manufacturer’s protocol. Cells were plated in a T150 flask with “monocyte media”, aMEM (Gibco, Grand Island, NY, USA) supplemented with 10% FBS, 1% penicillin/streptomycin, and an additional 2 mM L-Glutamine (Corning, Manassas, VA, USA), and incubated at 37 °C for 48 h. The non-adherent viable fraction of cells was collected and plated at 1 × 10^5^ cells/cm^2^ in 96-well plates with monocyte media supplemented with 30 ng/mL Macrophage Colony-Stimulating Factor (M-CSF) (Biolegend, San Diego, CA, USA). Plates were incubated for 48 h, after which the media was replaced with monocyte media supplemented with 30 ng/mL M-CSF and 25 ng/mL RANKL (Biolegend, San Diego, CA, USA). Media changes were performed daily until osteoclasts formed as confirmed by microscopy (three to four days after RANKL induction). Cells were fixed with 4% paraformaldehyde and stained for Tartrate Acid Resistant Phosphatase (TRAP) according to standard protocols. Images were obtained at 100× magnification with a Zeiss Laser Capture Microdissection microscope using Zeiss AxioVision 4.8 software (Carl Zeiss Microscopy LLC, Thornwood, NY, USA). Images were stitched using Zeiss ZEN Software v3.0 (Carl Zeiss Microscopy LLC, Pleasanton, CA, USA), and the center 10% of the well (an area of approximately 0.034 cm^2^) was analyzed for nuclei counting. Osteoclasts were defined as TRAP-positive cells with three or more nuclei and were counted and categorized as small (3–9 nuclei), medium (10–19 nuclei), or large (20+ nuclei) osteoclasts by three blinded observers.

### 2.9. RNAseq of Osteoclast Progenitor Cells

Femurs and tibias were dissected and bone marrow was flushed and filtered through a 70 µM cell strainer (Fisher Scientific, Waltham, MA, USA). RBC lysis was performed using Hybri-Max Red Blood Cell Lysing Buffer (Sigma-Aldrich, St. Louis, MO, USA) as per manufacturer’s protocol and viability confirmed using trypan blue exclusion. Cells were preincubated with anti-mouse CD16/CD32 (93) prior to staining for the following cell surface markers: CD3e-FITC (145-2C11), NK1.1-FITC (PK136), CD45R-FITC (RA3-6B2), CD11b-PE (M1/70), CD115-PE-Cy7 (AFS98), and CD117-APC (2B8) (Invitrogen, MA, USA). Experimental samples were depleted of FITC-positive cells using the EasySep Mouse FITC Positive Selection Kit II (STEMCELL Technologies, Cambridge, MA, USA) as per the manufacturer’s protocol. FITC-negative cells were collected and stained with Live/Dead Fixable Violet (Invitrogen, MA, USA) prior to cell sorting by the Duke Cancer Institute Flow Cytometry Shared Resource. CD3e^−^ NK1.1^−^ CD45R^−^ CD11b^lo/−^ CD115^+^ CD117^+^, defined as bone marrow monocyte progenitor cells with high osteoclastogenic potential [23,24] were collected and RNA was extracted using the RNeasy Mini Kit (Qiagen, Germantown, MD, USA) as per the manufacturer’s instructions. RNA samples were submitted to the Duke Center for Genomic and Computational Biology Genomic Analysis and Bioinformatics Core for RNAseq. To obtain sufficient RNA for analysis, the sorted cells from seven to nine mice were pooled per sample, and two wild-type and three *Grk3*^−/−^ samples were analyzed in the presented data.

### 2.10. Micro-Computed Tomography (µCT)

Wild-type and *Grk3*^−/−^ mice were euthanized as per Duke University IACUC-approved protocols. Femurs and tibias were dissected and transferred to 10% neutral buffered formalin (VWR, Suwanee, GA, USA) fozr two to three days, washed with PBS and stored in PBS until µCT images were obtained. Histomorphometry values were obtained with a Scanco VivaCT 80 Scanner (Scanco Medical AG, Switzerland) set to 55 kVp and 145 µM according to procedure published in [21]. Additional images of bones were obtained with a Nikon XT H 225 ST μCT machine (Nikon Metrology NV, Leuven, Belgium) at 180 kVp and 5 µM voxel size, and captured using Nikon Metrology’s Inspect-X software Version XT 5.4. After µCT, the bones were decalcified in 14% EDTA for 10–14 days, and paraffin embedded for histopathology.

### 2.11. TRAP Staining

A staining solution was made up of 0.2 M sodium acetate buffer, pH 4.7–5.0, containing 0.01 M tartaric acid and 2% naphthol AS-MX phosphate (Sigma-Aldrich, St. Louis, MO, USA) (dissolved at 20 mg/mL in ethylene glycol monoethyl ether). Fast Red Violet LB salt (Sigma-Aldrich, St. Louis, MO, USA) was added to the solution at a final concentration of 0.3 mM and warmed in a water bath to 37 °C. Slides with 5 μm FFPE sections, which were deparaffinized in xylene and rehydrated to water through a series of graded ethanols, were incubated in the staining solution for 1 h when reddish-purple staining was visible. Slides were then rinsed in water, counterstained with 0.08% Fast Green for 90 s and rinsed again in several changes of water. After air drying, slides were mounted with Permount (Fisher Scientific, Waltham, MA, USA).

### 2.12. Immunofluorescent Staining

Tissues were fixed with 4% paraformaldehyde for one to two hours, dehydrated in 30% sucrose overnight, then embedded in OCT without decalcification to preserve antigen activity. Sections were cut at 7 µm. Osterix (Abcam ab22552, Cambridge, MA, USA) and endomucin (Abcam ab106100, Cambridge, MA, USA) were stained as previously described [25].

### 2.13. Ki67 Staining

Standard immunohistochemistry protocols were followed according to published protocols [26]. Ki67 primary antibody (Cell Signaling Technology #12202S, Danvers, MA, USA) was diluted 1:400 in Background Snipper (Biocare Medical, Pacheco, CA, USA) and incubated overnight at 4 °C. Brightfield images were acquired at 10× magnification with Keyence BZ-X810 microscope.

### 2.14. β Galactosidase Staining

Primary WT and *Grk3*-deficient osteoclast precursors, obtained as described above, were plated at 1 × 10^5^ cells/cm^2^ in 12-well plates in monocyte media supplemented with 30 ng/mL M-CSF (Biolegend, San Diego, CA, USA). Plates were incubated for 48 h, after which the media was replaced with monocyte media supplemented with 30 ng/mL M-CSF and with or without 25 ng/mL RANKL (Biolegend, San Diego, CA, USA). Staining for senescence-associated β galactosidase was performed similar to published protocols [27] at timepoints 0, 1, 24, and 48 h after addition of RANKL using a senescence detection kit (Abcam 65351, Cambridge, MA, USA) according to manufacturer recommendation. Briefly, at the appropriate timepoint, the cells were washed and fixed. Staining was performed by incubation with 0.5 mL of freshly prepared X-gal staining solution mix at 37 °C for 1 h in a sealed zip-top bag to prevent CO_2_ exposure. The staining solution was removed, and the wells overlaid with a 70% glycerol solution and stored at −20 °C until image acquisition. Brightfield images were taken with a Keyence BZ-X810 microscope (Itasca, IL, USA) using a 10× objective for 4 random fields of each well from 4 independent experiments. These images were analyzed with ImageJ (U.S. National Institutes of Health, Bethesda, MD, USA) and the number of blue β galactosidase positive cells was expressed as a percentage of total cells.

### 2.15. Serum ELISAs

Blood was obtained from wild-type and *Grk3*^−/−^ mice using tail vein nicks at three month intervals. Blood was allowed to coagulate for two hours at room temperature, then spun down and serum was collected and frozen at −80 °C until analysis. ELISAs performed include bone-specific alkaline phosphatase (Biomatik, Kitchener, ON, Canada), TRAP (Lifeome BioLabs, Oceanside, CA, USA).

### 2.16. Expression of Grk3 in Human Peripheral Blood

Whole blood samples were collected into Paxgene tubes from 18 patients with PDB and 18 unaffected age-matched controls. Total RNA was isolated using Ambion Ribopure Blood RNA Purification Kit (AM1928) and RNA was quantified using the Nanodrop 1000 Spectrophotometer. Complementary DNA was generated by RT-PCR using the qScript cDNA SuperMix kit following the manufacturer’s instructions. Primers and fluorescently labelled probes were designed using the Roche Diagnostics website (Roche Diagnostics, Indianapolis, IN, USA). Primers used for qRT-PCR were: *Grk3* Human mRNA Forward: ttcagcgagacgttagcaag and *Grk3* Human mRNA Reverse: atgctgccagtcaacacctt. UPL Probe 26 (Sigma-Aldrich, St. Louis, MO, USA) was used for *Grk3* qPCR. Real-time PCR was performed on diluted cDNA using SensiFAST Probe No-ROX kit on a Bio RAD CFX Connect system and analyzed using the Bio RAD CFX Manager V1.0. *Grk3* expression were normalized to 18 s rRNA expression. 18 s cDNA was amplified with the VIC-labelled predesigned probe–primer combination from Applied Biosystems (4319413E, Waltham, MA, USA) allowing two channel detection of one cDNA.

### 2.17. Statistics

All data were graphed utilizing GraphPad Prism v.9 and statistically evaluated using GraphPad Prism v.9 (GraphPad, San Diego, CA, USA). Student’s *t*-test compared two independent groups (WT and *Grk3^−/−^*) when data were normally distributed, otherwise a Mann–Whitney U comparison was utilized. A two-way ANOVA was used when multiple comparisons were being analyzed. Analysis of *Grk3* mRNA expression in blood samples from cases and controls was evaluated by an unpaired *t*-test and chi square test.

## 3. Results

### 3.1. Aged Grk3^−/−^ Mice Have Abnormal Bone Lesions That Are Osteoclast Rich and Resemble PDB by µCT and Histopathology

Some diseases of bone remodeling occur more frequently with advanced age, such as osteoporosis and PDB. We have previously published that *Grk3^−/−^* BmMSCs have enhanced osteogenic differentiation and proliferation, whereas adipogenesis and chondrogenesis were similar to WT controls [21]. In light of this finding, we hypothesized that the *Grk3^−/−^* mouse would have a protective phenotype against osteoporosis and preserved bone density during aging. However, this was not found to be the case, and instead we observed that aged (17–20 month old) female *Grk3^−/−^* mice had similar trabecular bone morphometry (as measured by trabecular bone volume fraction on µCT) to WT female age-matched controls [21]. To determine whether or not further aging the mice or including male gender would elicit a phenotype, we expanded our cohort to include male and female mice aged 18–24 months. Although trabecular bone was similar to our previously published findings [21], we did observe changes in cortical bone (Figure 1). Aged *Grk3^−^*^/*−*^ mice had significantly decreased cortical bone thickness (Figure 1A) and increased bone surface to bone volume ratio (Figure 1C) when compared to age- and gender-matched WT mice. Although cortical bone volume was statistically different (Figure 1B), the presence of a notable WT outlier may have affected this result since all other WT and *Grk3^−^*^/*−*^ mice were similar in this measured parameter (Figure 1B).

Most notably, we observed striking lesions detected in the µCT scans of aged *Grk3*^−/−^ femurs that were not present in young *Grk3^−^*^/*−*^ mice (Appendix A) or age-matched WT mice (Appendix A and Figure 2A). To further define these lesions, we sectioned bone containing lesions that were detected on µCT and performed TRAP staining, which showed osteoclast enrichment within the lesions (Figure 2C,D) as well as staining for markers of bone remodeling. Specifically, the lesions exhibited disorganized neovascularization and osteoblast orientation as demonstrated by positive staining for endomucin (red) and osterix (green), respectively (Figure 2B).

To determine whether bone lesion penetrance was variable by sex, we examined µCT scans of femurs from male and female *Grk3*^−/−^ mice aged 22 months and found that both sexes have high levels of penetrance (87% and 91%, respectively), and that a vast majority of mice with one bone lesion have additional lesions (polyostotic disease) (83%) (Table 1).

### 3.2. Grk3 Expression Is Upregulated in Human and Mouse Cells during Osteoclastogenesis

Given that the aged *Grk3^−/−^* mouse had lesions resembling human PDB and that the primary defect in PDB is thought to reside in osteoclasts [2], we wanted to determine whether *Grk3* expression correlates with osteoclastogenesis. To investigate this, we queried publicly available gene expression data sets for *Grk3* expression in relevant cells/tissues (Table 2).

In both humans and mice, osteoclasts are formed by the fusion of mononuclear precursor cells which occurs after stimulation with M-CSF and RANKL [32]. As demonstrated from RNA expression data by Carey et al., *Grk3* RNA levels increase in human osteoclast precursor cells cultured with M-CSF and RANKL to induce osteoclastogenesis [28]. Additionally, human osteoclast precursors treated with bisphosphonates, a therapy for PDB, have sustained *Grk3* expression [29]. Similar to humans, murine bone marrow precursor cells cultured with M-CSF and RANKL show an elevation in *Grk3* RNA expression, and mouse osteoclasts highly express *Grk3* [30]. Additionally, the murine cell line RAW 264.7 has lower *Grk3* expression than primary murine cells [30].

Interestingly, an inactivating mutation in the intracellular motif IVVY of the RANK protein results in deficient RANKL signaling via the NFATc1 pathway. Cells with this altered domain showed completely abolished *Grk3* expression induction when treated with RANKL compared to WT controls [31].

Another study analyzed the expression profile of two transformed RAW 246.7 cell clones, one that retained the ability to form large osteoclasts in the presence of RANKL, while the other did not. *Grk3* expression was lower in the clone with defective osteoclastogenesis [33].

### 3.3. Grk3 Expression Is Decreased in Peripheral Blood Mononuclear Cells from PDB Patients

After determining from expression databases that GRK3 is important in osteoclast function, we wished to confirm whether our observations in the *Grk3^−/−^* mice were directly relevant to PDB patients. We therefore analyzed whole blood samples for *Grk3* expression in 18 PDB patients and 18 controls that were age- and gender-matched (Table 3). Since osteoclasts are tissue based, we analyzed gene expression from osteoclast hematopoietic precursors (monocytes) obtained from research subjects’ whole blood. Expression data showed that *Grk3* was significantly decreased in PDB patients (*p* = 0.016), which is analogous to *Grk3^−/−^* mice.

### 3.4. Grk3^−/−^ Osteoclast Progenitors Show Decreased Histone Gene Expression Compared to Controls

In an attempt to find a mechanistic explanation for the observed phenotype in *Grk3^−^*^/*−*^ mice, we performed RNAseq on a subset of bone marrow cells previously characterized as being a monocytic progenitor pool with high osteoclastogenic potential [23,24]. The population of interest was defined as CD3ε^−^ NK1.1^−^ CD45R^−^ CD11b^lo/−^ CD115^+^ CD117^+^ cells. In our analysis, RNAseq did not reveal obvious gene expression changes associated with osteoclastogenesis between groups (Figure 3). One WT mouse had noticeably different gene regulatory pathways activated, so heatmap analysis was performed both with (Figure 3A) and without (Figure 3B) the outlier, but this still did not elucidate a clear mechanistic pathway. We did note that several histone genes were comparatively downregulated in the *Grk3*^−/−^ CD3ε^−^ NK1.1^−^ CD45R^−^ CD11b^lo/−^ CD115^+^ CD117^+^ cells. Downregulation of histones has been associated with cellular senescence [34] and upregulation has been associated with a prolonged life-span [35]. Additionally, emerging data suggest roles for GPCR signaling in cellular senescence, a process by which aging cells develop progressively impaired replicative, secretory, and clearance mechanisms in response to cumulative cellular stress [36]. Consequently, we evaluated β galactosidase staining in cultures of WT and *Grk3*-deficient osteoclast precursors with and without RANKL to determine whether cellular senescence could be a mechanism explaining the PDB-like bone lesions. However, our in vitro differentiation data did not show differences at baseline or with RANKL in the number of cells positive for β galactosidase staining (Figure 4).

### 3.5. Grk3 Deficiency Increases Proliferation of Osteoclast Precursors, Which May Contribute to Increased Osteoclastogenesis

GRK3 has been linked to increased proliferation of several cell types through enhanced MAPK signaling [21,37,38]. We considered that increased numbers of osteoclasts seen in the PDB-like lesions of aged *Grk3*^−/−^ mice may be due to increased proliferation of monocyte/macrophage precursor cells since osteoclasts are terminally differentiated and do not proliferate. To directly investigate the role that *Grk3* may play in osteoclast precursor biology, we first used RAW 264.7-derived osteoclasts. RAW 264.7 are an Abelson leukemia virus-transformed cell line of mouse monocyte lineage that have been used as a model of osteoclastogenesis in vitro for over two decades [39]. First, we confirmed using qRT-PCR that *Grk3* expression is low/not detectable in RAW 264.7 cells (as seen in Table 1) but increases when RAW 264.7 cells are differentiated into osteoclasts by stimulation with RANKL (Figure 5A). Next, we transformed RAW 246.7 cells using shRNA lentiviral particles to knock down *Grk3* expression (hereafter termed “RAW *Grk3*-KD” cells). We evaluated *Grk3* expression in osteoclasts derived from the RAW *Grk3*-KD clones and selected two partial knockdown clones (700-2 and 701-2) and a control clone (NT-2) with *Grk3* expression intact for comparison experiments (Figure 5B).

To determine whether *Grk3* deficiency affects the number or size of osteoclasts, we performed flow cytometry of RAW *Grk3*-KD osteoclasts to quantify the total number of osteoclasts and to distinguish between different sizes of osteoclasts, based on the number of nuclei. RAW *Grk3*-KD clones and the control NT-2 clone were cultured in RANKL for six days and nuclear content was determined based on the previously reported flow cytometry procedure and gating strategy by Madel et al. [22]. The RAW *Grk3*-KD clones generated more osteoclasts overall compared to the control clone with intact *Grk3* expression, although this trend was not statistically significant (Figure 6).

While the trends of increased osteoclasts in the RAW *Grk3*-KD cultures were compelling, we considered that the lack of statistical significance in RAW-derived osteoclast numbers may have been reflective of residual *Grk3* expression that was not completely abrogated by shRNA knockdown (Figure 5B). To address this, we proceeded to investigate the effect of complete GRK3 deficiency on osteoclastogenesis using primary cells from *Grk3*^−/−^ mice [37,38]. When cultured with RANKL for 48 h, *Grk3^−^*^/*−*^ bone marrow-derived cultures generated significantly more osteoclasts compared to the WT cultures in total and within each of the number-of-nuclei size groups analyzed (Figure 7).

We have previously reported, using another methodology, that young *Grk3*^−/−^ mice have increased bone marrow cells and increased numbers and proliferation of immune cell subtypes in vitro [37]. To further investigate the role of increased proliferation of bone marrow-derived osteogenic precursors in vivo, we stained femurs isolated from both young (3 month) and old (18–22 month) *Grk3*-deficient and WT mice with immunohistochemistry for Ki67. Ki67 is a nuclear protein that is only expressed by proliferating, non-quiescent cells and as such is a marker of cellular proliferation [40]. Ki67 staining reproduced our previous findings of increased cellular proliferation of bone marrow niche cells [21,37] (Figure 8A) in young mice, but more striking was the increased Ki67 staining in *Grk3*^−/−^ aged mice (Figure 8B) relative to the similarly aged WT mice (Figure 8D).

### 3.6. Grk3 Deficiency Does Not Affect Resorption Capacity of Osteoclasts

After showing increased proliferation of osteogenic precursor cells and more numbers of osteoclasts generated in vitro, we wished to determine whether GRK3 deficiency affects the functional capacity of osteoclasts to resorb matrix, which is thought to be enhanced in human PDB. We performed a resorption assay with the RAW *Grk3*-KD clones, but GRK3 deficiency did not have an effect on resorption capacity in vitro when compared to WT control cells (Figure 9).

### 3.7. Serum Biomarkers of Bone Remodeling Associated with PDB Are Altered in Aged Grk3^−/−^ Mice

Given that the aged *Grk3^−^*^/*−*^ mice had marked abnormalities in bone histomorphometry (Figure 1) and osteolytic lesions (Figure 2) but similar resorptive capacity in vitro (Figure 9), we wanted to determine whether we could detect changes in serum biomarkers of bone remodeling in aged (18–24 month old) *Grk3^−^*^/*−*^ mice. Serum biomarkers are frequently used in the diagnosis and monitoring of metabolic bone diseases, including PDB [41,42,43], and serum biomarkers of osteoblast activity, such as total alkaline phosphatase (ALP) and bone-specific ALP (bALP), are good indicators of PDB disease activity and bisphosphonate treatment efficacy in humans [41,43]. Serum TRAP also positively correlates with biomarkers of bone formation, and thus disease activity, in PDB patients [42,44].

Interestingly, age- and sex-dependent differences were observed in serum bALP values in mice, but no differences were observed that were specific to genotype. (Figure 10A). Serum TRAP values were decreased in aged *Grk3^−^*^/*−*^ mice compared to WT mice, contrary to expectations (Figure 10B).

## 4. Discussion

To our knowledge, this is the first report describing a direct role for GRK3 in osteoclastogenesis in vivo. Several independent studies, including ours, confirm that *Grk3* expression increases during osteoclastogenesis in both human and mouse cells (Table 1). We show that deficiency of GRK3 in osteoclasts results in an altered phenotype with an increase in the number of osteoclasts generated in vitro in both a mouse cell line and primary bone marrow isolates (Figure 6 and Figure 7). We show that aged *Grk3^−/−^* mice exhibit osteoclast-rich lesions (Figure 2) analogous to those seen in humans with PDB and that marrow precursor cells maintain enhanced proliferation by Ki67 staining, whereas the resorptive function of osteoclasts does not appear to be affected, and serum biomarkers of bone turnover are not elevated.

Osteoclasts have been thought to be quiescent cells that typically do not replicate or undergo cellular senescence. However, recent data suggest osteoclasts could be senescent under certain conditions [45], and our RNAseq data showed some differences in histone gene expression between *Grk3*-deficient and WT mice. β galactosidase staining, which is widely used as a marker of cellular senescence in vivo and in vitro, was not different at baseline, over time, or between genotypes, making this particular mechanism less likely to explain the PDB-like lesions in our *Grk3*-deficient mice. The authors acknowledge limitations of bulk RNAseq data in that having to pre-select certain cell types for homogeneity may narrow the analysis such that other molecular pathways are missed [46]. It is possible that the subpopulation of osteoclasts that are abnormal in the *Grk3^−^*^/*−*^ mouse does not originate from this bone marrow population, as osteoclasts have several precursor cells [32].

GRK3 has been shown to be important in cellular migration [37,47]. This mechanism may be of relevance to PDB since increased osteoclasts observed within bone lesions in vivo could originate from precursors migrating from a bone marrow or vascular source. For example, we have shown previously that *Grk3* overexpression in a line of metastatic breast cancer cells decreased CXCR4 internalization and increased cellular migration and metastasis in a mouse model of breast cancer [47]. Additionally, we have shown that GRK3 modulates CXCR4-mediated migration [37], and CXCR4 expression is present on osteoclasts. However, when CXCR4 agonists or antagonists are added to culture media during osteoclastogenesis, the total number of osteoclasts generated is unaffected; therefore, we do not think that this is the mechanism of the observed osteoclast-rich lesions in the cortical bone of *Grk3^−/−^* mice. In previous work, we similarly showed that GRK3-mediated regulation of CXCR4 does not affect osteoblast proliferation [21], so this appears to be a similar finding in osteoclasts. The authors do acknowledge that CXCR4-mediated migratory effects could still be a relevant mechanism in vivo, which is the subject of future study; however, our Ki67 and in vitro data support enhanced proliferation of osteoclast precursors as the underlying mechanism.

Our *Grk3^−/−^* aged mouse shows several similar clinical features to other mouse models of PDB [11,14] and to humans, but it does not fully recapitulate the human disease. Although *Grk3* was not identified by GWAS, familial hereditary genetic studies and GWAS have identified only a handful of genes that explain ~20% of non-familial PDB cases [1,7,48], and there may be some genes that are dysregulated in PDB that were below the limit of detection in the human cohort studies. Our data support that regulatory defects in gene expression may also be important to mechanisms that underly PDB pathogenesis, as well as gene–environment interactions that could influence gene expression, and this could play a larger role in non-hereditary PDB [1,2]. Indeed, our findings of decreased gene expression of *Grk3* in human Paget’s patients (Table 3) would support this hypothesis.

Further understanding of the role of GRK3 in osteoclast biology may provide valuable information beyond PDB where osteoclasts play an important role in disease. Specifically, chronic periodontitis, osteoporosis, rheumatoid arthritis, and cancer metastases are exacerbated when osteoclasts are excessively recruited, inappropriately activated, or accumulate. Because the GPCR superfamily has been one of the more successfully targeted classes of receptors for drug therapy [49], GPCR antagonism of a receptor that GRK3 regulates, or direct kinase inhibition of GRK3 could be important for therapies in bone metabolism in the future.

## Figures and Tables

**Figure 1 cells-12-00981-f001:**
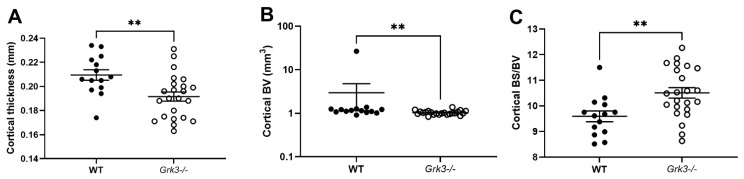
Aged 18–24 mo Grk3^−/−^ mice have altered cortical bone histomorphometry. (**A**) Cortical thickness; (**B**) Cortical bone volume (BV); (**C**) Cortical bone surface (BS)/BV ratio as measured by uCT, ** *p* < 0.01.

**Figure 2 cells-12-00981-f002:**
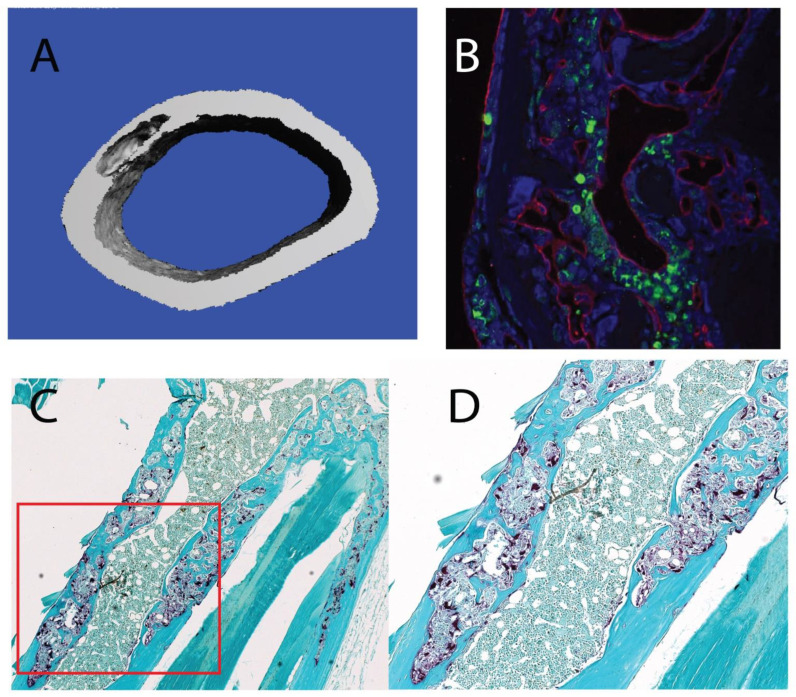
Aged 18–24 mo *Grk3^−/−^* mice have femur lesions detected on uCT that are osteoclast-rich and resemble pagetic lesions from human Paget’s disease of bone (PDB) patients and other PDB mouse models. (**A**) 3D reconstruction of a representative lesion seen on uCT from a 24 month *Grk3^−/−^* female. (**B**) Immunofluorescence of bone lesion stained for osteoblast marker osterix (green) and endothelial marker endumucin (red) with DAPI nuclear stain. (**C**) 5× magnification of an immunohistochemistry-stained lesion positive for TRAP (tartrate resistant alkaline phosphatase) osteoclast marker, and (**D**) 40× higher power magnification shows disorganized cellular architecture within cortical bone that has a mosaic appearance consistent with PDB.

**Figure 3 cells-12-00981-f003:**
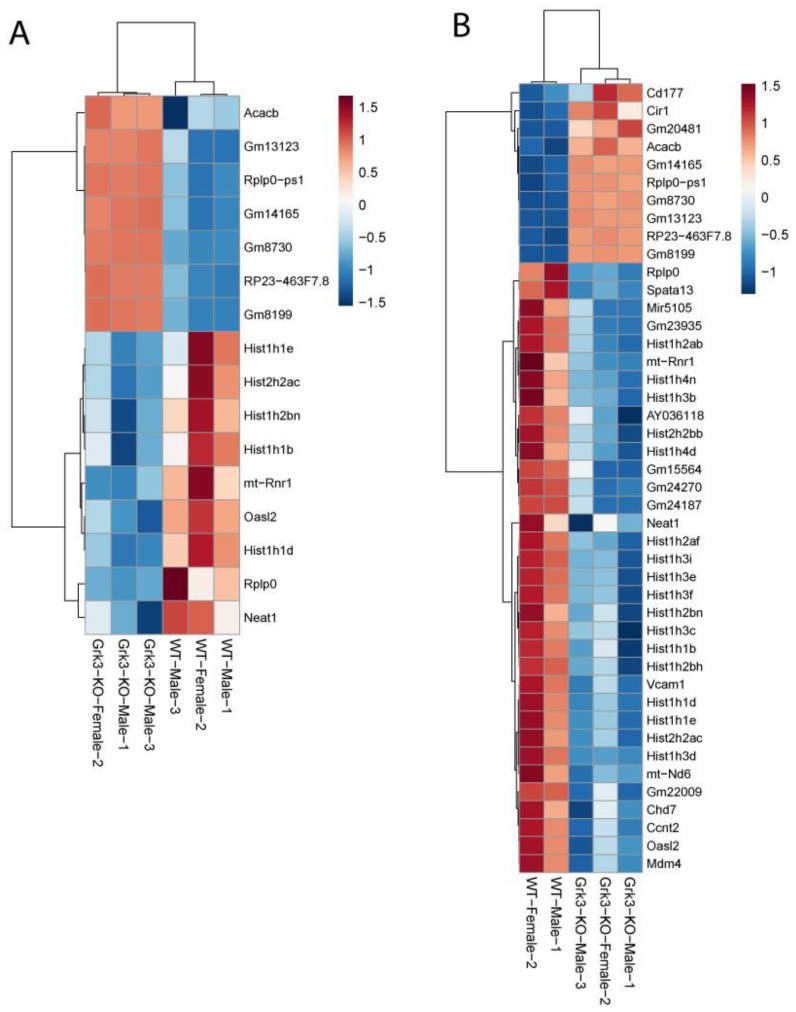
RNAseq gene expression of *Grk3^−/−^* and wildtype bone marrow derived pre-osteoclasts sorted by FACS as CD3ε- NK1.1- CD45R- CD11blo/- CD115+ CD117+ cells. (**A**) Bone marrow from 3 mice of the same age/gender/genotype were pooled, and monocyte progenitor cells previously characterized as having high osteoclastogenic potential were purified by FACS according to published techniques [22,23] for RNAseq analysis. (**B**) RNAseq analysis repeated with one WT outlier removed.

**Figure 4 cells-12-00981-f004:**
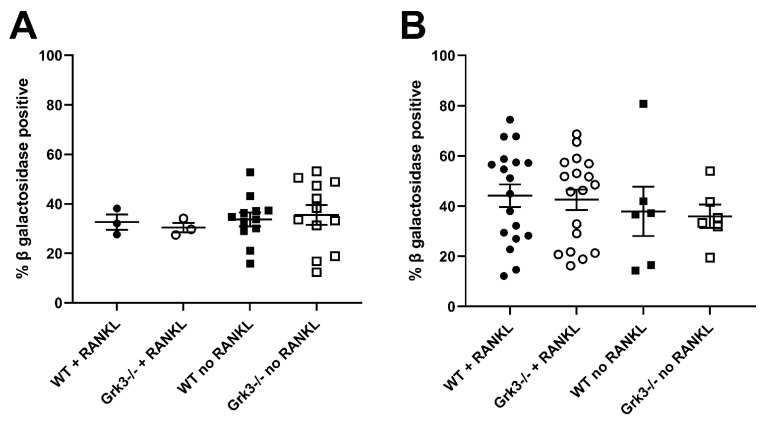
β galactosidase staining of cultured primary bone marrow derived monocytes from *Grk3^−/−^* and wildtype mice at (**A**). baseline (0 h) and at (**B**). 48 h in the presence or absence of RANKL.

**Figure 5 cells-12-00981-f005:**
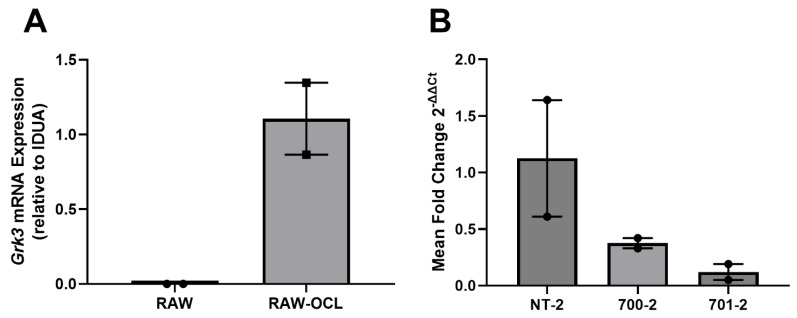
*Grk3* expression in RAW 264.7 cells during osteoclastogenesis and after shRNA knockdown. (**A**) *Grk3* is not detectable in cultured RAW 264.7 cells in regular growth media but expression is induced when RAW cells are treated with RANKL to induce osteoclastogenesis as confirmed by TRAP staining. (**B**) RAW 264.7 shRNA knockdown of *Grk3* leads to substantial reduction of *Grk3* expression in two control lines compared to NT-2 nonspecific lentiviral control.

**Figure 6 cells-12-00981-f006:**
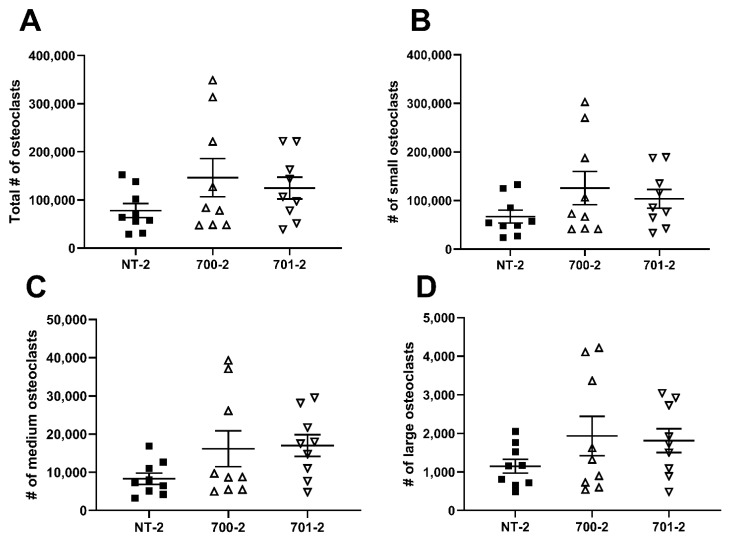
*Grk3*-deficient RAW cells generate more osteocalsts in vitro. Flow cytometric enumeration of osteoclasts based on nuclei number; (**A**) total osteoclasts (>3 nuclei), (**B**) small osteoclasts (3–4 nuclei), (**C**) medium osteoclasts (5–8 nuclei) and (**D**) large osteoclasts (9+ nuclei) after differentiation in RANKL [22].

**Figure 7 cells-12-00981-f007:**
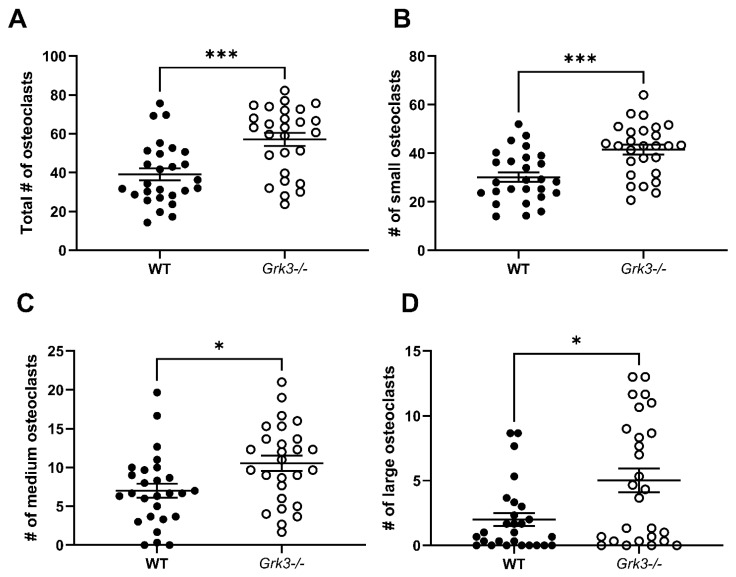
Primary bone marrow-derived precursors from *Grk3^−/−^* mice generate more osteoclasts ex vivo. (**A**) Total osteoclasts (>3 nuclei), (**B**) small osteoclasts (3–4 nuclei), (**C**) medium osteoclasts (5–8 nuclei) and (**D**) large osteoclasts (9+ nuclei) were significantly increased (* *p* < 0.5 and *** *p* < 0.001) after differentiation in RANKL, staining for TRAP, and enumeration.

**Figure 8 cells-12-00981-f008:**
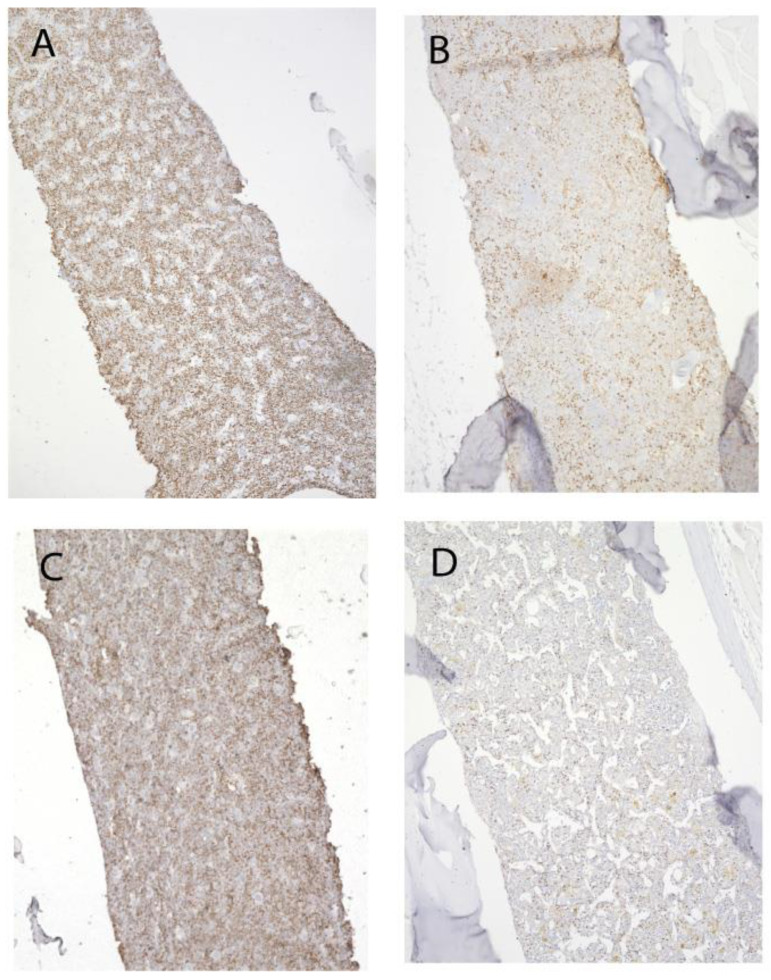
Proliferation of bone marrow cells remains enhanced in advanced aging of *Grk3^−^/^−^* mice. Representative Ki67 staining of (**A**). *Grk3^−/−^* mouse 3 months old, (**B**) *Grk3^−/−^* mouse 22 months old, (**C**) WT mouse 3 months old, and (**D**) WT mouse 18 months old.

**Figure 9 cells-12-00981-f009:**
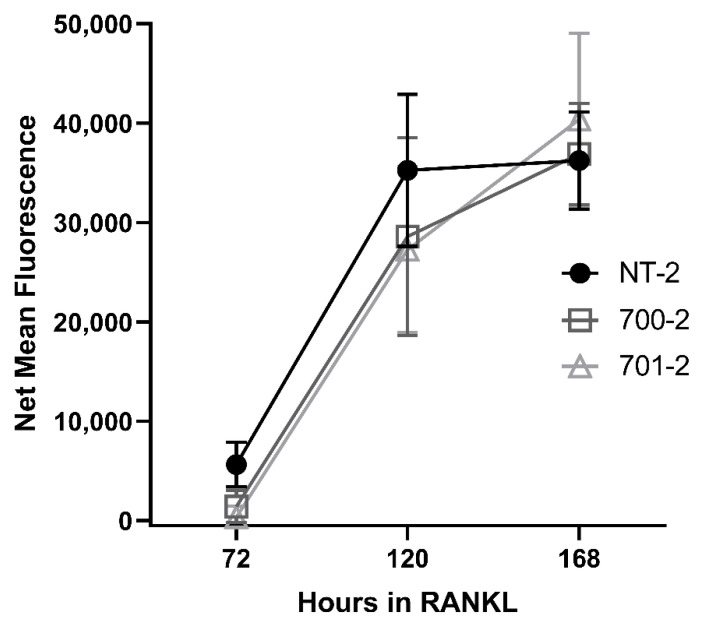
*Grk3* deficiency does not increase resorption capacity of osteoclasts in vitro. After differentiation into osteoclasts with RANKL, two *Grk3* knockdown RAW264.7 cell lines show similar mineral resorption to the shRNA knockdown control cell line.

**Figure 10 cells-12-00981-f010:**
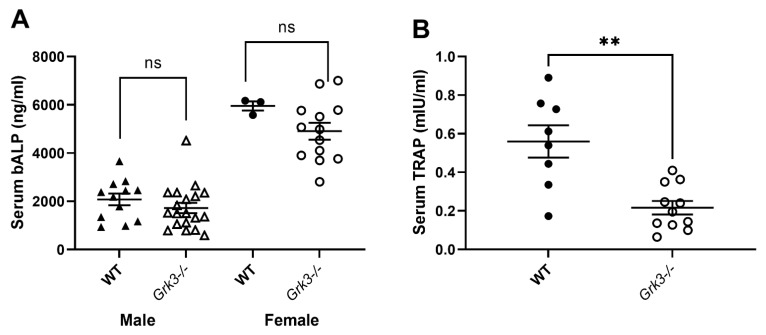
Serum TRAP but not bone alkaline phosphatase (bALP) are different in aged (18–24 month) *Grk3^−/−^* mice compared to controls. (**A**) Female mice have increased bALP compared to males, but there was no significant (ns) difference between WT and *Grk3*-deficient mice when gender was controlled. (**B**) Serum TRAP was significantly decreased (** *p* < 0.01) in *Grk3^−/−^* mice.

**Table 1 cells-12-00981-t001:** Incidence and distribution of bone lesions are not affected by sex in aged *Grk3^−/−^* mice.

Aged *Grk3^−/−^* Mice	At Least One Lesion Present	Of Those with One Lesion: More Than One Lesion Present
Male	87% (13/15)	69% (9/13)
Female	91% (10/11)	100% (10/10)
Total	88% (23/26)	83% (19/23)

**Table 2 cells-12-00981-t002:** *Grk3* gene expression in osteoclastogenesis and osteoclast-targeted therapy.

Publication	GEO Dataset	Cell Type	Experimental Condition	*Grk3* Expression
[28]	GSE107295	Human osteoclast precursor cell	Cultured with M-CSF and RANKL to induce osteoclast formation	*Grk3* increases with RANKL treatment
[29]	GSE63009	Human peripheral blood cell derived osteoclast precursors	Cultured in M-CSF and RANKL then treated with bisphosphonates	*Grk3* expression is sustained in osteoclast precursor cells treated with bisphosphonates
[28]	GSE107295	Mouse bone marrow derived myeloid precursor cells	Cultured with M-CSF +/− RANKL	*Grk3* expression increases with RANKL treatment
[30]	GSE10246	Variety of murine tissues		Higher expression of *Grk3* in osteoclast precursor cells than other cells in the bone
[30]	GSE10246	RAW 264.7 cells		*Grk3* expression is lower than primary murine bone cell types
[31]	GSE30160	Mouse bone marrow macrophages with inactivating motif on RANK domain	Culture with M-CSF and RANKL	*Grk3* expression induction during osteoclastogenesis is abolished with inactivating mutation in the RANK IVVY domain

**Table 3 cells-12-00981-t003:** *Grk3* expression in PDB patients and controls.

	PDB (*n* = 18)	Controls (*n* = 18)	*p*-Value
Age (years)	75.6 ± 12.0	73.8 ± 13.6	0.89
Female	13 (72%)	15 (83%)	0.69
Grk3/18 s (AU)	0.83 ± 1.14	2.28 ± 3.11	0.016

Values are mean ± SD. The *p*-values refer to differences between groups assessed by *t*-test or chi-square test.

## Data Availability

The data presented in this study are available on request from the corresponding author.

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
