# Peer review of "Aged G Protein-Coupled Receptor Kinase 3 (Grk3)-Deficient Mice Exhibit Enhanced Osteoclastogenesis and Develop Bone Lesions Analogous to Human Paget’s Disease of Bone"

_cells, 2023, doi:10.3390/cells12070981_

Round 1
Reviewer 1 Report
The manuscript by Rabjohns et al gives an intriguing insight into the potential role of Grk3 in PDB, via effects on osteoclasts, which result in PDB-like lesions in Grk3-KO mice.
Although the authors were unable to define a mechanism whereby Grk3 affects osteoclastogenesis, the authors present a thorough preliminary investigation of the effects of Grk3 in this regard, both in vitro and in vivo.
I have only a few minor points:
Introduction: Can the authors specify what protein is OPTN?
Methods – Please describe the source of the antibodies used for osterix and endomucin staining.
Figure 1 – It looks as though the significance of the cortical BV microCT is driven entirely by the one outlier? This seems an obvious outlier, with all other animals the same (in both groups)…
Figure 3 legend – panels are described as 4A and 4B
Author Response
In response to Reviewer 1, we have clarified the following within the manuscript (see revised version attached).
- The optineurin protein (OPTN) is now clarified and defined in the Introduction in lines 72 and 76.
- The vendor manufacturer for endomucin and osterix antibodies has been added to Methods in lines 237-238.
- Reviewer 1 comments on there being an outlier that may have led to the statistical significance of Figure 1B (Cortical BV). For experimental rigor, we did not omit this animal from data analysis, particularly since its other parameters fell within range of the cohort for other variables. However, we agree that this should be commented on, and we have added this commentary in Results lines 306-308.
- Figure 3 typographical errors mislabeling panels 4A and 4B are corrected in lines 682 and 684.
Reviewer 2 Report
The manuscript describes the involvement of GRK3 in the regulation of osteoclast differentiation and shows that aged mice deficient in GRK3 expression have bone lesions similar to those developed by human PDB. It is well written, the methods are carefully described, the results are clearly presented. It deserves to be published, after some minor revision.
The heatmaps shown in Figure 3A and B should be organized in the same way, i.e. on the left WT-Male-1 and WT-Female-2 …. Grk3-KO-Male -1 … or vice versa, to make the visualization of the results clearer and for a more immediate comparison between A and B.
The legends of Figures 3, 4, 5, 6, 7, 8, 10, must be carefully checked for some typing errors: see for example “Figure 3. RNAseq gene expression of…... 4A. Bone marrow …. 4B. RNAseq analysis …
Author Response
In response to Reviewer 2, we have clarified the following within the manuscript:
- We agree that the heatmaps in Figure 3A and B should be organized in the same way, but unfortunately, the Core Facility that generated these data has closed, and we do not have access to the software to regenerate the figure. We apologize for this; however, the data are accurate, and we felt it was of value to show the data presented with the outlier removed so that readers could draw their own conclusions about the genetic data. We hope that you agree.
- Thank you. for bringing to attention the typographical error in the legend of Figure 3. We have addressed this in lines 682 and 684 as well as proofread the manuscript again for other errors.